# Mechanochemistry Frees Thiourea Dioxide (TDO) from the ‘Veils’ of Solvent, Exposing All Its Reactivity

**DOI:** 10.3390/molecules28052239

**Published:** 2023-02-28

**Authors:** Francesco Basoccu, Federico Cuccu, Pietro Caboni, Lidia De Luca, Andrea Porcheddu

**Affiliations:** 1Department of Chemical and Geological Sciences, University of Cagliari, 09042 Monserrato, Italy; 2Department of Chemical, Physical, Mathematical, and Natural Sciences, University of Sassari, Via Vienna 2, 07100 Sassari, Italy

**Keywords:** heterocycles, thiourea dioxide, TDO, mechanochemistry

## Abstract

The synthesis of nitrogen-based heterocycles has always been considered essential in developing pharmaceuticals in medicine and agriculture. This explains why various synthetic approaches have been proposed in recent decades. However performing as methods, they often imply harsh conditions or the employment of toxic solvents and dangerous reagents. Mechanochemistry is undoubtedly one of the most promising technologies currently used for reducing any possible environmental impact, addressing the worldwide interest in counteracting environmental pollution. Following this line, we propose a new mechanochemical protocol for synthesizing various heterocyclic classes by exploiting thiourea dioxide (TDO)’s reducing proprieties and electrophilic nature. Simultaneously exploiting the low cost of a component of the textile industry such as TDO and all the advantages brought by a green technique such as mechanochemistry, we plot a route towards a more sustainable and eco-friendly methodology for preparing heterocyclic moieties.

## 1. Introduction

Heterocycles are ubiquitous in biologically active compounds, natural products, and common pharmaceuticals [1,2,3,4,5,6,7,8,9], representing a highly privileged structural motif. For example, common biocides [10], fungicides [11], antitumoral agents [12,13], and analgesics [14,15,16], to mention a few, contain a benzimidazole or benzothiazole moiety in their structure. Furthermore, the benzimidazole scaffold represents a benchmark for synthesizing new potential agents against various cancer or infectious diseases, even those that still cannot be effectively treated [17,18]. This is due to the benzimidazole ring’s astonishing proprieties that simultaneously possess a hydrophobic unit and two hydrogen-bonding domains (Figure 1) [19].

Accordingly, several endeavors have been dedicated to the synthesis of such compounds. Among all the reported methods, those using either formaldehyde or an appropriate surrogate reagent have proven to be efficient and valuable choices for preparing different heterocycles containing the benzimidazole core [20,21,22,23,24,25,26]. However, these strategies usually suffer from some limitations, such as employing additives, e.g., metal catalysts, harsh reaction conditions, or toxic solvents (Figure 1).

In this context, thiourea dioxide (TDO), a solid surrogate for formaldehyde, could allow us to overcome many of the shortcomings mentioned above. TDO is a cheap and commercial compound commonly employed in textile industries for bleaching processes [27]. Moreover, it has also been exploited in the past for analytical measurements [28,29,30]. Finally, it can also be synthesized, as pioneered by Barnett (Figure 2) [31], Lai [32], and Kluttz [33], and different studies have already been published concerning both its electrophilic and reducing nature [34,35,36,37,38,39].

As a matter of fact, TDO is insoluble in most organic solvents as well as in water, and its reactivity can be triggered by raising the temperature. Because of this, it is typically used in mixtures of methanol and basic or heated water [40,41].

In 2021, Wu’s team documented benzimidazoles’ synthesis through thiourea dioxide’s (TDO, **1**) electrophilic character in a solvent-based process [42]. The procedure was carried out in water at 60 °C, providing around ten benzimidazoles in satisfactory yields. A solvent screening was also made on the control reaction, proving how the insolubility of TDO in organic solvents made it unreactive compared to the employment of hot water. In addition, the use and removal of solvents during a chemical synthesis represents a significant portion of organic pollution and process energy consumption.

Benzimidazoles are generally synthesized from aniline derivatives, which in turn are often prepared from the corresponding nitrobenzenes. Furthermore, the synthetic methods to produce anilines still involve classical methodologies, and there is currently a worldwide interest in proposing new synthetic routes to improve the sustainability and applicability of such a process [43,44,45,46]. Reducing an aromatic nitro moiety usually requires an acid environment in the presence of a metal [47,48,49,50] or the employment of gaseous hydrogen [51,52,53,54]. Despite performing satisfactorily, these methodologies have concerns that must be addressed. Starting from metals, this procedure is usually run under an acidic environment [54,55,56,57,58], and consequently, it may not be applied to substrates sensitive to such conditions.

Furthermore, the use of metals is often associated with various risks concerning human health and the environment due to their well-established toxicity. For example, with molecular gaseous hydrogen, its use is related to an explosion hazard because of its high reactivity [55,56]. Since this procedure is exergonic (ΔrG° < 0), a high amount of heat is typically released during an industrial process [57]. Such heat generation can be challenging to remove and may also induce expensive cooling costs or lower the reaction yields due to reactor hot spots [58,59]. Other methods rely on electrochemistry [59,60], enzymatic processes [56], rare-earth elements [61], and hydrogen transfer reagents [61,62,63,64], all of which need specific reaction conditions and advanced equipment. Few reductions in a basic environment have been described in literature [65,66,67]; the most recurrent one is undoubtedly the employment of Zn powder in the presence of NaOH for synthesizing azobenzene [68]. Unfortunately, a basic environment generally implies other sub-products formed during the redox process [57,67,69]. Their presence complicates the obtention of the corresponding anilines, making the entire procedure complex and cumbersome.

On the contrary, the reducing processes of TDO are associated with the release of nontoxic side products, mainly urea and sodium sulfite [37].

The abovementioned drawbacks prompted us to assess the feasibility of grinding the reaction under solvent-free conditions. Indeed, ball-milling remains an impressive technology in this regard [69,70,71,72]. Many mechanochemical strategies described astonishing advantages such as high reaction efficiency, the prevention of harsh reaction conditions, and the minimization of organic solvents [73,74,75].

As part of our ongoing interest in green synthesis via mechanochemistry [47,48], we evaluated whether heat could be easily replaced by mechanical energy, thus enabling access to higher and more sophisticated reactivities.

## 2. Results and Discussion

### 2.1. TDO as a Reducing Agent

Firstly, the reducing properties of TDO (Compound **1**) on nitrobenzenes were explored to outline and screen all the mechanochemical parameters for this step. Considering the few existing techniques for reducing nitrobenzene with TDO [76,77], we had to lay the groundwork for a methodology with a broader applicability. We set the mechanochemical procedure on a 1.0 mmol scale using nitrobenzene as a reference substrate. We milled TDO **1** (1.0 mmol), nitrobenzene **2a** (1.0 mmol), and NaOH (1.0 mmol) for 1.0 h inside a 10 mL stainless steel (SS) vessel equipped with two balls (Ø = 7 mm, 2.67 g) of the same material (Figure 3).

Unfortunately, we detected the only presence of the starting material through a GC-MS analysis (Table 1, entry 1). Several process parameters have been investigated to overcome these failures, and the whole optimization process is summarized in Table 1. To begin, we raised the ratio of **1** and NaOH (Table 1, entries 2 and 3), which allowed us to convert **2a** into aniline **3a** with a 21% yield (Table 1, entry 3). Then, prompted by these results, we tried to increase both the reaction time and the reducing mixture equivalents (TDO and NaOH). In the first case, the yield was even lower in a 2 h milling with a 5% yield (GC-analysis), probably due to the higher reactivity of the formed aniline with the redox intermediates (Table 1, entry 4). In the latter one, **2a** was consumed entirely, but the conversion to the desired product **3a** increased slightly together with other process intermediates (Table 1, entry 5). Lastly, we ran the reducing process at 70 °C to accelerate the kinetics of the reaction, but we only obtained the corresponding symmetric diazobenzene (PhN=NPh, Table 1, entry 6).

Therefore, after all these failed attempts, we considered using drops of different solvents to run a **L**iquid-**A**ssisted **G**rinding (**LAG**) [78,79,80,81,82] in a 90 min procedure. Consistently, less polar (decane or toluene) and polar solvents (acetone or isopropanol) did not permit a reasonable conversion rate (Table 1, entries 7–10). Lastly, methanol and water were used, as in analogous solvent-based procedures. However, in this case, the ratio of solvent/reagents was drastically cut down compared to the already reported methodologies (LAG, η = 0.44 µL/mg). In contrast to that which is usually found for solvent-based procedures, methanol used under LAG conditions produced a complex mixture of aniline and nitrobenzene reduction process intermediates (Table 1, entry 11). Water, instead, led to excellent yields of **3a** (Table 1, entry 12). Its amount, however, was found to be a critical parameter, since the conversion rate dramatically dropped when η = 0.22 µL/mg (Table 1, entry 13). Contrarily, a little increase in the reaction time of up to 2 h allowed for a quantitative conversion of **2a** to **3a** (Table 1, entry 14). Concerning the bases, weaker ones such as sodium carbonate and sodium bicarbonate did not allow for a comparable result (Table 1, entries 15 and 16), proving that NaOH plays a crucial role in the mechanochemical process—consistent with the findings of Hawkes for similar reactions in solution [83]. Lowering the NaOH amount negatively affected the reaction performance as well (Table 1, entry 17).

With the optimized conditions in hand, we extended the entire procedure to other nitrobenzenes to validate this mechanochemical process. In the case of activated substrates **2b** and **2c**, the process smoothly proceeded to a complete conversion within 2 h, and for more complex substrates like **2s**, the process was completed in only 3 h (Figure 4).

Reducing instead 2-nitroaniline **2d**, many unpredicted outcomes showed up, as summarized in Table 2. In this case, we synthesized the *o*-phenylenediamine **3d** with a 37% yield in 90 min without LAG (Table 2, entry 1). Such different behavior can be ascribed to the positive effects of an EDG. Unexpectedly, prolonging the reaction time to 2 h under neat grinding conditions resulted in the formation of the corresponding benzimidazole, albeit in low yields (Table 2, entry 2). For the sake of completeness, we also tried to reduce in neat conditions other substrates having a comparable charge distribution (Table 2, entries 3 and 4). With 2-nitro anisole, we obtained the corresponding aniline in a 54% isolated yield. At the same time, the employment of 2-nitro phenol resulted in a mixture of various unidentified products, likely generated by the high phenoxide reactivity.

To better understand several critical details of the process, we have to better focus on several points of the process. First, under LAG conditions, the presence of methanol resulted in the concurrent formation of **3d** and various reaction intermediates. At the same time, water use was associated with the synthesis of the desired product with a nearly quantitative yield in 2 h (Table 2, entries 5 and 6). These different outcomes can be reconducted to the role covered by water as a better proton source. On the other hand, water, to some extent, also inhibits the final cyclization pathway that leads to benzimidazole formation. The reduction in the nitro group and the construction of the benzimidazole ring both consume TDO, with the latter being kinetically faster. As a result, any attempt to decrease the NaOH equivalents failed because of the high reactivity of the formed *o*-phenylenediamine towards TDO that was still present in the reaction medium (Table 2, entry 7). Once these issues are focused on, we can draw conclusions based on the above and after a long, meticulous exploratory study. Six equivalents of NaOH promote the sluggish kinetics of the reduction reaction to the detriment of the cyclization reaction, resulting in a complete reduction of the nitro group (Table 2, entry 6).

Once we understood the reactivity of 2-nitroanilines, we also extended this process to other 2-nitroaniline derivatives (Figure 5). As a result, *o*-phenylenediamines **3e** and **3f** were successfully synthesized in a 2 h ongoing process, whereas the substrates **3g**–**j** and **3o**–**p** needed a longer reaction time of 3 h. These outcomes were utterly in line with Hammett’s parameters and steric hindrance on the aromatic ring of the starting materials.

### 2.2. TDO as an Electrophile

Having clarified the role of TDO **1** as a reducing agent, we thoroughly analyzed its application as a “green” solid replacement for formaldehyde for synthesizing aza-heterocycles. Its electrophilicity is connected to the presence of the two nitrogen atoms depleting the central carbon atom in terms of electron density. Furthermore, an excellent leaving group such as the sulfur moiety makes the whole molecule more prone to a nucleophilic attack. We accomplished the heterocycle synthesis by establishing two separate procedures: a single-step process based on the employment of phenylenediamines (**procedure A**) and a double-step methodology starting from 2-nitroanilines (**procedure B**).

#### 2.2.1. Procedure A

The reactivity of phenylenediamines toward compound **1** has already been reported in the literature for solvent-based processes restricted to a few very reactive substrates. [42]. We investigated the mechanosynthesis of heterocycles from less reactive substrates, in this case, the ones that possess a lower electron density in their aromatic ring (Figure 6). All the reactions were conducted using a LAG, where water was used as the additive (η = 0.44). Starting from the *o*-phenylenediamines **3l**–**n**, we obtained the corresponding heterocycles in low yields due to the low reactivity of such compounds. Deactivating groups either on the ring or on the nitrogen atom drastically affected the ring closure process, as evidenced by the poor yields obtained for compounds **4l**–**n**. Remarkably, diamines **3q** and **3r** enabled access to molecular framework of biological interest in poor to good yields [84,85,86].

#### 2.2.2. Procedure B

With the optimal conditions for reducing 2-nitroanilines in hand, we wondered whether a double-step methodology could be feasible. Bearing in mind that a basic environment consumes compound **1** for forming the reducing species, we realized that we needed to add an additional amount of it for running the second step. In addition, we added a small quantity of water, because the process was proved to perform poorly through neat grinding, as formerly stated. After these considerations, we shaped the procedure to convert 2-nitroanilines into aza-heterocycles (Figure 7). The first step was run under fine-tuned conditions, so the newly formed *o*-phenylenediamine was ready for the forthcoming ring closure step. This last stage was successfully accomplished with a refill of **1** (3.0 mmol) and water (250 μL, η = 0.44), and it was run for further 2 h for the substrates **2d**–**f**. The 2-nitroanilines **2g**–**k** and **2o**–**p** required a longer reaction time of 3 h instead. 

The reaction proceeded well in all the considered cases and provided benzimidazoles **4d**–**4j** in near-quantitative yields except for compounds **4k** and **4p** (Figure 7).

To assess the green footprint of this mechanochemical procedure, we calculated the green metrics for our methodology and compared them with a previously reported solvent-based process. The results highlight a substantial improvement, in a green chemistry framework, of the proposed mechanochemical technique concerning the solvent-based approach (see the paragraph “Green Metrics” in the Appendix A for further details).

To further demonstrate the potentialities of the developed mechanochemical protocol, various trials on a larger-scale reaction (from 2 mmol up to 5 mmol) were conducted. As illustrated in Figure 8, the mechanochemical solvent-free reaction of **2a** (5.0 mmol) with **1** (30.0 mmol), NaOH (30.0 mmol), and water (1.5 mL) at 30 Hz for 4 h gave product **4a** in a satisfactory product yield of 91%, showing the robustness of the present method and how it could be feasibly adapted for a possible scale-up process.

All the syntheses presented are easy to accomplish and proceed with a first redox process followed by a final ring closure and aromatization step (Figure 9). The reducing ability of compound 1 has been widely described, and it exploits the formation of sulfoxylic acid [36,37,38,87,88]. However, such a chemical species can be formed only from the tautomer of compound 1, aminoiminomethanesulfinic acid (AIMS), usually only observed in an aqueous medium (Figure 9, pathway 1) [89,90,91,92]. After being formed, it is converted to its more stable form, sodium hydrogen sulfoxylate, by the remaining equivalents of NaOH (Figure 9, pathway 2). It is hard to imagine that a species such as sodium sulfoxylate could be formed due to the low acidity of sodium hydrogen sulfoxylate [87]. This last compound is then ready to participate in the redox process and will be reduced to gaseous sulfur dioxide (Figure 9, pathway 3).

Nonetheless, we cannot completely rule out the presence of sodium dithionite and sodium bisulfite as reducing agents. The former can be formed by TDO degradation [39,93], and the sulfur dioxide may generate the latter in the presence of water [94]. Ultimately, adding fresh **TDO** to the reaction medium allowed the ring to be closed to the corresponding aromatic heterocycle (Figure 9, pathway 4).

Concerning water (250 μL) used to perform the LAG process (η = 0.44), it plays three fundamental roles in the mechanochemical redox reaction. Firstly, it enables the formation of the active tautomer AIMS**,** as described by Dittmer [91] and Krug [92]. Secondly, the complete consumption of **1** to sodium sulphoxylate in a strong basic environment avoids other collateral processes such as the auto condensation of TDO to cyclic derivatives, as reported in the literature [95]. Thirdly, the presence of water might prevent the formation of other undesired redox intermediates, because it acts as a proton donor [96]. Lastly, it probably permits sodium hydroxide to participate extensively in concert with TDO due to its high solubility in water. This may also explain the poorer performance of the mechanochemical process when methanol is employed for a LAG approach.

To conclude, we also studied the reactivity of hydrazobenzene under our optimized conditions. Unfortunately, conducting the reaction with 1 mmol of hydrazobenzene with TDO (3 mmol), NaOH (6 mmol), and water (250 μL) only yielded the starting material as described by Huang [77]. Therefore, the entire redox process is wholly described in Figure 10.

Considering the ring closure step, the electrophilic nature intrinsic to compound **1** makes possible the formation of a heterocycle (Figure 11a,b). We firmly support the idea that, in the presence of NaOH, cyanimide cannot be formed by the degradation of compound **1** [97]. Therefore, the only possible pathway for heterocycle synthesis is the release of formamidine through a dismutative process (Figure 11a pathway 1) [98,99]. After undergoing a nucleophilic attack from the *o*-phenylenediamine, formamidine allows the generation of an *N*-arylformamidine intermediate that will go through a second nucleophilic attack from the other nitrogen atom (Figure 11a, pathway 2). The resulting 2-amino dehydrobenzimidazole then extrudes the NH_2_ moiety as ammonia for forming the benzimidazole structure (Figure 11a, pathway 3).

Nevertheless, we cannot rule out a reaction mechanism based on the supposed carbenoid structure of TDO as well [89,100,101,102] (Figure 11b). In this case, the reaction mechanism initially follows a diverse pathway based on the minor charge separation between the two moieties of TDO (Figure 11b, pathway 1) [90,103]. Then, after forming an *N*-arylformamidine intermediate, the process continues as mentioned above (Figure 11b, pathway 1). 

We ruled out a possible deamination approach from the corresponding 2-aminobenzimidazole for the following reasons. Firstly, when we milled commercial 2-aminobenzimidazole in the presence of NaOH, we did not see any change in the starting material’s nature. Secondly, the thermodynamic stability of such a substrate prevents any structure alteration under our mild conditions [104]. Thirdly, its synthesis should be associated with an unlikely dehydrogenative process in our reaction medium. Finally, once the entire process is finished, the desired product needs to be extracted from the reaction mixture.

Along with the newly synthesized heterocycle, other subproducts were formed. We think their presence was due to the degradative processes of compound **1**, as already documented [95,105,106]. Hence, a short silica pad was made to obtain the desired product in high purity.

## 3. Materials and Methods

### 3.1. Materials

Commercially available reagents were purchased from Acros (Geel, Belgium), Aldrich (Darmstadt, Germania), Strem Chemicals (Newburyport, MA, USA), Alfa-Aesar (Haverhill, MA, USA), and TCI Europe (Zwijndrecht, Belgium) and used as received. All of the reactions were monitored by thin-layer chromatography (TLC) performed on glass-backed silica gel 60 F254, 0.2 mm plates (Merck, Darmstadt, Germania), and compounds were visualized under UV light (254 nm) or using cerium ammonium molybdate solution with subsequent heating. The eluents were technical grade. The mechanochemical reactions were performed using a Retsch Mixer Mill MM 500 VARIO apparatus (horizontal vibratory mill). The reagents were milled using a stainless steel grinding jar (10 mL) equipped with two balls (Ø = 7.00 mm, 2.67 g) of the same material. The ^1^H- and ^13^C-NMR spectra were recorded on a Bruker (Billerica, MA, USA) Avance III HD 600 MHz NMR spectrometer at 298 K. Proton chemical shifts are expressed in parts per million (ppm, δ scale) and are referred to as the residual hydrogen in the solvent (CDCl_3_, 7.27 ppm or DMSO-*d*_6_ 2.54 ppm). Carbon chemical shifts are expressed in parts per million (ppm, δ scale) and are referenced to the carbon resonances of the NMR solvent (CDCl_3_, 77.0 ppm or DMSO-*d*_6_ 39.5 ppm). GC-MS analyses were performed on an Agilent 5977B MS interfaced to the GC 7890B equipped with a DB-5ms column (J & W, New Brighton, UK). Yields refer to pure, isolated materials.

### 3.2. General Procedure A for Anilines and o-Phenylenediamines ***3a–j***, ***3o–p***, ***3s*** Synthesis from 2-Nitroanilines ***2a–j***, ***2o–p***, ***2s***

A 10 mL stainless steel jar equipped with two stainless steel milling balls (7 mm diameter, 2.67 g) was filled with nitrobenzenes **2a**–**j, 2o**–**p, 2s** (1.0 mmol), NaOH (6.0 mmol), **1** (3.0 mmol), and 250 μL of distilled water. The vessel was then closed, and the mechanochemical reaction was conducted, ranging from 60 to 180 min at 30 Hz. Whenever necessary, further purification through flash column chromatography was performed. Lastly, the solvent was removed under reduced pressure to afford the pure anilines **3a**–**j, 3o**–**p, 3s**. 

### 3.3. General Procedure B for Heterocycles ***4l**–**n***, ***4q**–**r*** Synthesis from o-Phenylenediamines ***3l**–**n***, ***3q**–**r***

A 10 mL stainless steel jar equipped with two stainless steel milling balls (7 mm diameter, 2.67 g) was filled with o-phenylenediamines **3l**–**n**, **3q**–**r** (1.00 mmol), 1 (2.00 mmol), and 200 μL of distilled water. The vessel was then closed, and the mechanochemical reaction was conducted, ranging from 60 min to 180 min at a frequency of 30 Hz. At the end of the reaction, an additional silica pad (SiO_2_, heptane/ethyl acetate/methanol = 1:1:0→6:3:1) was made to purify the reaction mixture. Lastly, the solvent was removed under reduced pressure to afford the pure heterocycle **4l**–**n**, **4q**–**r**.

### 3.4. General Procedure C for Heterocycles ***4d**–**k***, ***4p*** Synthesis from 2-nitroanilines ***2d**–**k***, ***2p***

A 10 mL stainless steel jar equipped with two stainless steel milling balls (7 mm diameter, 2.67 g) was filled with 2-nitroanilines **2d**–**k**, **2p** (1.0 mmol), NaOH (6.0 mmol), 1 (3.0 mmol), and 250 μL of distilled water. The vessel was then closed, and the mechanochemical reaction was conducted, ranging from 60 to 180 min at 30 Hz. After that, an additional refill of **1** (3.0 mmol) was made, and 60 μL of distilled water and the mechanochemical reaction was made to run ranging from 60 to 180 min at a frequency of 30 Hz. At the end of the reaction, an additional silica pad (SiO2, heptane/ethyl acetate/methanol = 1:1:0→6:3:1) was made to purify the reaction mixture. Lastly, the solvent was removed under reduced pressure to afford the pure heterocycle **4d**–**k**, **4p**.

## 4. Conclusions

This work has thoroughly explained a mechanochemical protocol for synthesizing heterocycles, rediscovering a solid reagent as thiourea dioxide (TDO). Not only did the dual nature of such a compound allow us to propose both reducing and ring closure procedures, but it also allowed us to merge these two processes in a one-pot technique starting from 2-nitroanilines. By avoiding a mixture of methanol and basic water like in the in-solution methods, we were also able to deeply analyze the already-known reducing properties of TDO, as has never been reported. The reaction is easy to perform and allows for the obtainment of the desired products with yields ranging from low to excellent. In addition, this methodology provided an alternative pathway for synthesizing scaffolds of biological and pharmaceutical interest, such as benzimidazole derivatives, and valuable building blocks with a potential application in drug design, such as perimidines and imidazopyridines.

## Data Availability

The data presented in this study are available in the Appendix A.

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
