# Peer review of "Mechanochemistry Frees Thiourea Dioxide (TDO) from the ‘Veils’ of Solvent, Exposing All Its Reactivity"

_molecules, 2023, doi:10.3390/molecules28052239_

Round 1

Reviewer 1 Report

This manuscript describes the mechanochemical use of thiourea dioxide (TDO) to synthesize various heterocyclic compounds, which, as the authors note, are of broad usefulness in the pharmaceutical and agricultural industries. Descriptions are supplied of experimental variations employed to optimize reaction yields, including the nature and quantity of added bases, the use of liquid-assisted grinding, and the length of grinding time. There is nothing truly novel in the heterocycles that are produced—instead, a focus is on the reduction of solvent use and the milder reaction conditions employed, which are typical benefits of mechanochemically driven reactions. The audience for the paper would seem to be synthetic organic chemists and possibly those interested in green/sustainable chemistry. After addressing some issues, the paper could be suitable for publication.

1) The novelty of the chemistry is somewhat overstated. The authors describe TDO as “an old forgotten solid reagent” (page 14, line 396), but they mention various uses in the Introduction, some relatively recent. There are other applications than those they cite (e.g., Mendonça, P. V., et al., “Thiourea Dioxide As a Green and Affordable Reducing Agent for the ARGET ATRP of Acrylates, Methacrylates, Styrene, Acrylonitrile, and Vinyl Chloride.” ACS Macro Letters (2019), 8, 315-319). They state that they were able “to analyze the reducing properties of TDO, as nobody has ever reported” (page 14, line 401), but TDO’s use in the formation of heterocycles is not new (see: Verma, S.; Kumar, S.; Jain, S. L.; Sain, B. “Thiourea Dioxide Promoted Efficient Organocatalytic One-Pot Synthesis of a Library of Novel Heterocyclic Compounds,” Org. Biomol. Chem. (2011), 20, 6943- 6948; this reference should be added to the manuscript). (Ironically, the authors of that paper comment on how the low solubility of TDO is actually an advantage in their chemistry: “Thiourea dioxide is found to be insoluble in various organic solvents and therefore at the end of the reaction products can be separated by extraction with diethyl ether and the recovered catalyst can be used several times with consistent catalytic activity.”) This is not to say that the present manuscript does not offer previously unreported findings. Still, these lie mainly in mechanochemical activation, not the reducing properties of TDO.

1) Although I am usually reluctant to criticize titles, to suggest that mechanochemistry lifts the “veils” of solvents from a reaction misstates the fundamental nature of mechanochemically driven reactions. It implies an underlying reaction mechanism that solvent use obscures and that the true nature of reagent interactions will be revealed by grinding/milling. There is no guarantee that a solid-state mechanochemical reaction will occur by the same route as a solution-based reaction—reagents are at higher concentrations than in solution, for example, and conditions are often far from equilibrium. I am sure that the authors are aware that the investigation of reaction mechanisms is currently a ‘hot’ area of mechanochemical research.

Simply saying that mechanochemical reaction conditions allow TDO to display reactivity not usually observed in solution would state the nature of the paper more concisely.

2) Yields are not always described consistently. For example, the products obtained from the o-phenylenediamines 3l-n are said to have been obtained in “poor yield” (page 7, line 232), and Scheme 6 specifies that these ranged from 10%–25%. In contrast, the diamines from 3q and 3r are said to have been obtained in “modest to good yields.” The diamine from 3q (i.e., 4q) was obtained in 11% yield; why is this described as “modest” whereas that for 4l, at 25%, is considered “poor”?

3) As the authors make clear in the Introduction, improved synthesis of heterocyclic compounds could be of substantial importance to industry, but this depends on whether the reactions can be efficiently scaled up. This is often a limitation of mechanochemical reactions, which might work well on a 100 mg scale, for example, but not on a gram scale—perhaps as a result of inadequate mixing during the grinding, excessive heat production (solvents help disperse heat!), or other experimental conditions. If possible, it might be helpful to redetermine the outcome of one or two of the more efficient reactions on a larger scale (e.g., by 5× or 10×) and see whether the yields remain roughly the same.

Usage/Typos/Clarifications needed

Title: The word “the” should be removed before “thiourea dioxide” and possibly before “solvent” (see also comments above about rewriting the title completely)

Abstract, lines 12,13: “However, performing as methods often imply harsh conditions or the employment of toxic solvents and dangerous reagents.” What does this sentence mean? Did the authors intend something like: “However, reactions performed in solution often require harsh conditions…”?

Abstract, line 16: “renewed” should be replaced with “new”

“However, performing as methods often imply harsh conditions or the employment of toxic solvents and dangerous reagents.” What does this sentence mean? Should it be: “However, reactions performed in solution often require harsh conditions…”

Page 1, line 27: “agents” should be inserted after “antitumoral”

Page 1, line 33: the phrase “proper citation of hydrogen bond donor and acceptor should be given” should be removed

Page 2, lines 38, 39: the sentence beginning “Among all…” would be smoother as “Among all the reported methods, those using either formaldehyde or an appropriate surrogate reagent have proven to be efficient and valuable choices for preparing different heterocycles containing the benzimidazole core”

Page 3, line 53; why are the studies “out of date”? The term implies something no longer valid or relevant—why would such studies no longer be valid? Even if “dated” is meant, the most recent publication cited is from 2016.; it would be best to remove the phrase “albeit out of date” altogether.

Page 3, lines 64,65: “…providing around ten benzimidazoles in satisfactory yields and a solvent screening on the control reaction proving the inefficacy of TDO in organic solvents due to a solubility matter.”  If the benzimidazoles were obtained in satisfactory yield, how does this prove the inefficacy of TDO?

Page 3, lines 75,76: “Despite entirely performing as methods, all the concerns regarding these methodologies must be remarked.”  What does this sentence mean? Did the authors intend something like: “Despite performing satisfactorily, these methodologies have concerns that must be addressed.”?

Page 3, line 67: “represent” should be “represents”

Page 4, line 138: “Contrasting what is known,” would be better as “In contrast to what is usually found…”

Page 4, line 146: “…as already insight by Hawkes for similar reactions in solution [83].” would be better as “…consistent with the findings of Hawkes for similar reactions in solution [83].”

Page 6, line 170: “without a LAG” should just be “without LAG”

Page 7, lines 215,216: “Clarified the role of TDO 1 as a reducing agent, we thoroughly analyzed its application as a green solid surrogate of formaldehyde…” could perhaps be improved with something like: “Having clarified the role of TDO 1 as a reducing agent, we thoroughly analyzed its application as a “green” solid replacement for formaldehyde…”

Page 7, line 232: “for compound 3l-n” should be changed to “for compounds 4l-n” or “from compounds 3l-n”

Page 8, line 242: “specie” should be “species” (“specie” means money in the form of coins rather than notes). This error occurs repeatedly in the manuscript, and the authors should do a simple find-and-replace to correct them all.

Page 8, line 243: “…we needed a refill of it (fresh adding) for running the second step.” would be better as “…we needed to add an additional amount of it for running the second step.”

Author Response

Reviewer 1:
This manuscript describes the mechanochemical use of thiourea dioxide (TDO) to synthesize various heterocyclic compounds, which, as the authors note, are of broad usefulness in the pharmaceutical and agricultural industries. Descriptions are supplied of experimental variations employed to optimize reaction yields, including the nature and quantity of added bases, the use of liquid-assisted grinding, and the length of grinding time. There is nothing truly novel in the heterocycles that are produced—instead, a focus is on the reduction of solvent use and the milder reaction conditions employed, which are typical benefits of mechanochemically driven reactions. The audience for the paper would seem to be synthetic organic chemists and possibly those interested in green/sustainable chemistry. After addressing some issues, the paper could be suitable for publication.
1) The novelty of the chemistry is somewhat overstated. The authors describe TDO as “an old forgotten solid reagent” (page 14, line 396), but they mention various uses in the Introduction, some relatively recent. There are other applications than those they cite (e.g., Mendonça, P. V., et al., “Thiourea Dioxide As a Green and Affordable Reducing Agent for the ARGET ATRP of Acrylates, Methacrylates, Styrene, Acrylonitrile, and Vinyl Chloride.” ACS Macro Letters (2019), 8, 315-319). They state that they were able “to analyze the reducing properties of TDO, as nobody has ever reported” (page 14, line 401), but TDO’s use in the formation of heterocycles is not new (see: Verma, S.; Kumar, S.; Jain, S. L.; Sain, B. “Thiourea Dioxide Promoted Efficient Organocatalytic One-Pot Synthesis of a Library of Novel Heterocyclic Compounds,” Org. Biomol. Chem. (2011), 20, 6943- 6948; this reference should be added to the manuscript). (Ironically, the authors of that paper comment on how the low solubility of TDO is actually an advantage in their chemistry: “Thiourea dioxide is found to be insoluble in various organic solvents and therefore at the end of the reaction products can be separated by extraction with diethyl ether and the recovered catalyst can be used several times with consistent catalytic activity.”) This is not to say that the present manuscript does not offer previously unreported findings. Still, these lie mainly in mechanochemical activation, not the reducing properties of TDO.
Authors’ answer: the authors are grateful to this referee for his precious revision. We agree with his suggestion concerning the several uses of TDO, and we decided to replace “an old forgotten solid reagent” (page 14, line 396) with “a solid reagent” for this reason. Regarding the statement “to analyze the reducing properties of TDO, as nobody has ever reported” (page 14, line 401), we intended to highlight the lack of procedures to reduce nitrobenzenes with TDO and under a basic environment. To the best of our knowledge, the few articles reporting this are the following:
- Huang, S.-L.; Chen, T.-Y. Reduction of Organic Compounds with Thiourea Dioxide II. The Reduction of Organic Nitrogen Compounds. J. Chin. Chem. Soc. 1975, 22, 91-94, doi: https://doi.org/10.1002/jccs.197500011.
- Krug, P. Thiourea Dioxide (Formamidinesulphinic Acid) A New Reducing Agent for Textile Printing. J. Soc. Dye. 1953, 69, 606-611, doi: https://doi.org/10.1111/j.1478-4408.1953.tb02803.x.
- Nakagawa, K.M., S.; Kawamura, S.; Minami, K. . Reduction of Organic Compounds with Thiourea Dioxide. II Reduction of Aromatic Nitro Compounds and Synthesis of Hydrazo Compounds. Yakugaku Zasshi 1977, 97, 1253-1256.
All these articles merely describe the reduction of nitrobenzene to aniline and its corresponding azoderivatives without any attempt to expand the reaction scope.
Moreover, we also gave unprecedented insight into how the LAG conditions play a crucial role in the entire process, as demonstrated by the poor performances of neat grinding (Table 1, Entries 1-6). Concerning the paper by Sain et al., we see the referee’s point of view related to the reported use of TDO in the heterogeneous phase. Still, we respectfully disagree with the comparison made with our mechanochemical work. In the mentioned paper, TDO acts as an organocatalyst in synthesizing different heterocycles. Its insolubility in organic solvents enables excellent advantages in terms of its recovery and reusability. Instead, in documented in-solution procedures, TDO is used in a stoichiometric amount or even with a 3-fold excess. This means that all the compounds in the reaction mixture must be dissolved by using a combination of basic water and an alcoholic solvent or by heating the mixture. Consequently, we decided to modify our statement: "we were also able to deeply analyze the already known reducing properties of TDO as nobody has ever reported.” Lastly, the reference has been added (now ref. 9) as requested by the referee.
2) Although I am usually reluctant to criticize titles, to suggest that mechanochemistry lifts the “veils” of solvents from a reaction misstates the fundamental nature of mechanochemically driven reactions. It implies an underlying reaction mechanism that solvent use obscures and that the true nature of reagent interactions will be revealed by grinding/milling. There is no guarantee that a solid-state mechanochemical reaction will occur by the same route as a solution-based reaction—reagents are at higher concentrations than in solution, for example, and conditions are often far from equilibrium. I am sure the authors are aware that the investigation of reaction mechanisms is currently a ‘hot’ area of mechanochemical research.
Simply saying that mechanochemical reaction conditions allow TDO to display reactivity not usually observed in solution would state the nature of the paper more concisely.
Authors’ answer:
We fully understand the concerns raised by the reviewer, and often we have to debate the titles of our papers. Here our Italian roots emerge strongly, and with the utmost respect for the excellent work done by the reviewer, we like to argue it as if it was an open letter with him/her. We are aware that many mechanistic features witnessed in mechanochemical processes need to be fully understood, and often, we are unequipped to account for some experimental results. In this paper, we have opened a window on a process developed to date in solution, highlighting, as far as we could, the key role water plays as a solvent. We still need to clarify every aspect of the process exhaustively. We are also aware that the mechanochemistry process may follow pathways other than what is observed in the solution. Therefore, throughout the sections of the manuscript, we have left room for further studies on this subject, which is only a starting point. Of course, we are scientists first and foremost and have to be critical, first and foremost, with ourselves.
So, what is the point of such a strong title? To clarify these aspects, let us provide two examples/methaphors.
When the veil is picked up on artwork, we can begin to understand the painting or sculpture better. It takes years and years of study to understand an artwork, and often something escapes and calls into question what has been written before. Removing the veil opens up a long path of research and reflection. On the contrary, the veil's physical presence makes all analysis and meditation impossible.
In the same way, removing the veil on a crime scene will help understand several elements, but more clues and a lengthy police investigation are needed.
A headline attracts the reader's interest, driving his curiosity to get to the end of reading the text ... if it is of value. If the editor and referee agree, we trust them and keep the original title. Otherwise, we are ready to change it into: “Mechanochemistry allows a better understanding of thiourea dioxide (TDO) reactivity and properties,” in agreement with what was stated by the referee.
3) Yields are only sometimes described consistently. For example, the products obtained from the o-phenylenediamines 3l-n are said to have been obtained in “poor yield” (page 7, line 232), and Scheme 6 specifies that these ranged from 10%–25%. In contrast, the diamines from 3q and 3r are said to have been obtained in “modest to good yields.” The diamine from 3q (i.e., 4q) was obtained in 11% yield; why is this described as “modest” whereas that for 4l, at 25%, is considered “poor”?
Authors’ answer: Per the referee's report, we changed the yields of 3q and 3r from “modest to good” to “poor to good.”
4) As the authors make clear in the Introduction, improved synthesis of heterocyclic compounds could be of substantial importance to industry, but this depends on whether the reactions can be efficiently scaled up. This is often a limitation of mechanochemical reactions, which might work well on a 100 mg scale, for example, but not on a gram scale—perhaps as a result of inadequate mixing during the grinding, excessive heat production (solvents help disperse heat!), or other experimental conditions. It might be helpful to redetermine the outcome of one or two of the more efficient reactions on a larger scale (e.g., by 5× or 10×) and see whether the yields remain roughly the same.
Authors’ answer: The scale-up procedure was implemented in lines 264-269.
“To further demonstrate the potentialities of the developed mechanochemical protocol, various trials on a larger-scale reaction (from 2 mmol up to 5 mmol) were conducted. As illustrated in Scheme 8, the mechanochemical solvent-free reaction of 2a (5.0 mmol) with 1 (30.00mmol), NaOH (30.0 mmol), and water (1.5 mL) at 30 Hz for 4 h gave product 4a in a satisfactory product yield of 91%, showing the robustness of the present method and how it could be feasibly adapted for a possible scale-up process.”
Usage/Typos/Clarifications needed
Title: The word “the” should be removed before “thiourea dioxide” and possibly before “solvent” (see also comments above about rewriting the title completely)
Authors’ answer: The redundant presence of “the” has been modified as requested.
Abstract, lines 12,13: “However, performing as methods often imply harsh conditions or the employment of toxic solvents and dangerous reagents.” What does this sentence mean? Did the authors intend something like: “However, reactions performed in solution often require harsh conditions…”?
Authors’ answer: The sentence has been modified to make it clearer.
Abstract, line 16: “renewed” should be replaced with “new.”
“However, performing as methods often imply harsh conditions or the employment of toxic solvents and dangerous reagents.” What does this sentence mean? Should it be: “However, reactions performed in solution often require harsh conditions….”
Authors’ answer: The word “new” has been implemented, and the sentence has been modified to make it clearer.
Page 1, line 27: “agents” should be inserted after “antitumoral”
Authors’ answer: The term was added as requested.
Page 1, line 33: the phrase “proper citation of hydrogen bond donor and acceptor should be given” should be removed.
Authors’ answer: The phrase was removed as requested.
Page 2, lines 38, 39: the sentence beginning “Among all…” would be smoother as “Among all the reported methods, those using either formaldehyde or an appropriate surrogate reagent have proven to be efficient and valuable choices for preparing different heterocycles containing the benzimidazole core”
Authors’ answer: The entire phrase has been rewritten as requested.
Page 3, line 53; why are the studies “out of date”? The term implies something no longer valid or relevant—why would such studies no longer be useful? Even if “dated” is meant, the most recent publication cited is 2016.; it would be best to remove the phrase “albeit out of date” altogether.
Authors’ answer: Any incorrect statement about the value of previous studies has been canceled.
Page 3, lines 64,65: “…providing around ten benzimidazoles in satisfactory yields and a solvent screening on the control reaction proving the inefficacy of TDO in organic solvents due to a solubility matter.” If the benzimidazoles were obtained in satisfactory yield, how does this prove the inefficacy of TDO?
Authors’ answer: The inefficacy mentioned concerns the reported screening of solvents made by Wu et al., where it can be seen how the performance of their process in organic solvents is somewhat poorer compared to the results obtained in a hot water solution. To make the clause more thorough, we modified it.
Page 3, lines 75,76: “Despite entirely performing as methods, all the concerns regarding these methodologies must be remarked.” What does this sentence mean? Did the authors intend something like: “Despite performing satisfactorily, these methodologies have concerns that must be addressed.”?
Authors’ answer: The referee's suggestion has been integrated into the text.
Page 3, line 67: “represent” should be “represents”
Authors’ answer: The verb has been corrected as requested.
Page 4, line 138: “Contrasting what is known” would be better as “In contrast to what is usually found…”
Authors’ answer: The referee's suggestion has been integrated into the text.
Page 4, line 146: “…as already insight by Hawkes for similar reactions in solution [83].” would be better as “…consistent with the findings of Hawkes for similar reactions in solution [83].”
Authors’ answer: The referee's suggestion has been integrated into the text.
Page 6, line 170: “without a LAG” should just be “without LAG”
Authors’ answer: The expression was corrected as requested.
Page 7, lines 215,216: “Clarified the role of TDO 1 as a reducing agent, we thoroughly analyzed its application as a green solid surrogate of formaldehyde…” could perhaps be improved with something like: “Having clarified the role of TDO 1 as a reducing agent, we thoroughly analyzed its application as a “green” solid replacement for formaldehyde…”
Authors’ answer: The clause has been modified as requested.
Page 7, line 232: “for compound 3l-n” should be changed to “for compounds 4l-n” or “from compounds 3l-n”.
Authors’ answer: The clause was changed from “for compound 3l-n” to “for compounds 4l-n”.
Page 8, line 242: “specie” should be “species” (“specie” means money in the form of coins rather than notes). This error occurs repeatedly in the manuscript, and the authors should do a simple find-and-replace to correct them all.
Authors’ answer: The mistake was corrected in the entire manuscript.
Page 8, line 243: “…we needed a refill of it (fresh adding) for running the second step.” would be better as “…we needed to add an additional amount of it for running the second step.”
Authors’ answer: The sentence was corrected as requested.

Reviewer 2 Report

The authors presented avery well conducted study on the one pot; two pot (and also two steps one-pot) mechanochemical synthesis of reduction of o-nitroanilines and cyclization to nitrogen based heterocycles. In that purpose they reborn the thiourea dioxide solid compound that used it elegantly for both steps. The important point also is the LG necessary presence of water.

A series of o-nitroanilines have been reduced in th epresence of base to the corresponding orthophenylene diamine systems in very efficient manner.

Then the authors transformed very efficiently the diamines to the corresponding aza heterocycles. As the eraction conditions do not differ drastically between the two reactions, the authors also efficiently developed the two steps/one pot procedure.

The mechanistic hypotheses are sound. All compounds are well characterized.

In that respect , i find the manuscript very suitable for publication in Molecules.

Two minor remarks:

english corrections: line 91: obtention instead of obtainment. Line 94 On the contrary instead of contrarily.

A minor question: does th ereaction works with the commercially available urea:H202 system?

Author Response

differ drastically between the two reactions, the authors also efficiently developed the two steps/one pot procedure.
The mechanistic hypotheses are sound. All compounds are well characterized.
In that respect, I find the manuscript very suitable for publication in Molecules.
Authors’ answer: the authors are grateful to this referee for the revision. We managed to accomplish their requests concerning both the English language and the scientific trials.
Two minor remarks:
English corrections
Line 91: obtention instead of obtainment.
Authors’ answer: the word has been substituted as requested.
Line 94 On the contrary instead of contrarily.
Authors’ answer: the term has been modified as requested.
A minor question: does the reaction work with the commercially available urea:H2O2 system?
The authors thank the reviewer for this valuable suggestion. Unfortunately, urea hydroperoxide (UHP) was tested in the model reaction without any useful results. This finding was partly expected since the two reagents, TDO and UHP, have diametrically opposite behavior, reducing the former reagent and oxidizing the latter.
